# Analysis of the Composite Boring Bar Dynamic Characteristics Considering Shear Deformation and Rotational Inertia

**Chunjin Zhang [1],\*, Yongsheng Ren [1], Shujuan Ji [2] and Jinfeng Zhang [1]**

[1] College of Mechanical and Electronic Engineering, Shandong University of Science and Technology, Qingdao 266590, China; renys@sdust.edu.cn (Y.R.); zhangjf@sdust.edu.cn (J.Z.)

[2] College of Computer Science and Engineering, Shandong University of Science and Technology, Qingdao 266590, China; jane_ji@sdust.edu.cn

\* Correspondence: zhangchunjin@sdust.edu.cn; Tel.: +86-134-6828-0849

**Abstract:** A boring bar is a tool used to install cutters and transfer power during boring. Because the boring bar in a narrow workspace is usually slender, chatter often occurs in the boring process. To improve the chatter stability of the boring bar, researchers have designed composite material boring bars with high dynamic stiffness to meet the requirements of high-speed boring. However, the effect of the shear deformation and rotational inertia were ignored. In this paper, a model of a composite boring bar considering shear deformation and rotational inertia is established based on the Adomian Modified Decomposition Method (AMDM). The dynamic characteristics, such as the vibration mode shapes, natural frequency, and chatter stability of the composite boring bar considering the shear deformation and rotational inertia, are analyzed comprehensively. The analysis results show that, when the shear deformation and rotational inertia are considered, the composite boring bar can exhibit different vibration mode shapes. Moreover, the natural frequencies and the cutting depth will be reduced. The results are helpful to improve the understanding about dynamic characteristics of the composite boring bar, and to provide guidance for designing of boring bars. Moreover, accurate adjustment of the cutting speed and depth in CNC boring can be based on the analysis results.

**Keywords:** composite boring bar; Adomian Modified Decomposition Method (AMDM); chatter stability; shear deformation

## 1. Introduction

A boring bar is a tool used for installing cutters and transferring power during boring operation. Cutting holes with satisfactory depths is a difficult problem in boring work. When cutting deep holes, as the boring bar is slender and with a weak dynamic stiffness, chatter occurs easily. When chatter occurs, the surface finish of the cut will be poor, the cutting efficiency will be lowered, and the cutter may be destroyed. To overcome the problem of chatter and to effectively improve the chatter stability during boring, researchers have conducted many studies [1–10]. Based on the different methods of vibration reduction and control of a boring bar, these studies can be divided into the following three categories.

The first category is the design of dynamic vibration absorbers (DVAs). Many researchers have designed DVAs to reduce vibrations and improve the stability of boring bars. For example, Liu et al. [11] designed a new type of damping boring bar with a DVA. The test results showed that it could effectively reduce vibrations and render good cutting performances. Lee et al. [12] designed and analyzed a three-dimensional model of a damped boring bar using the ANSYS software. The results showed that the proposed damping boring bar exhibited good damping performance. Rubio et al. [13] focused on

the optimal selection of the parameters of a passive DVA attached to a boring bar. The results showed that their method yielded good improvements of the stability performance. Li et al. [14] designed a device to reduce vibrations by adding a DVA to the boring bar. By changing the axial pressure, their method could change the stiffness of the DVA and effectively suppress the vibrations of the boring bar. Liu et al. [15] designed a variable-stiffness dynamic vibration absorber (VSDVA), and the proposed dynamic model could produce the best vibration reduction by adjusting the stiffness of the VSDVA. Lei et al. [16] proposed a composite DVA with a particle damper (PD) to reduce the additional mass of the DVA and improve the vibration control performance. The results indicated that the vibration reduction performance of the composite DVA significantly outperformed a traditional DVA and single PD. The previous work showed that to design an ideal DVA, it is necessary to accurately understand the dynamic cutting parameters and cutting environment of the boring bar.

The second category is the design of active vibration reduction controllers. Many active vibration reduction controllers are designed to achieve active vibration reduction of the boring bar and achieve stable cutting. For example, Yigit et al. [17] studied the effect of piezoelectric shunt damping on the chatter vibrations in a boring process. Theoretical and experimental analysis showed that the application of piezoelectric shunt damping could achieve a significant increase in the absolute stability limit in boring operations. Lu et al. [18] presented a noncontact magnetic actuator fit with fiber optic displacement sensors. By mounting the actuator on a computer numerical control (CNC) lathe, they conducted experiments. The experimental results showed that the bending mode of the boring bar was actively damped, and the dynamic stiffness of the system increased significantly. To increase the damping and static stiffness of the boring bar during cutting, Chen et al. [19,20] designed a 3-degree freedom linear magnetic actuator. Cutting tests showed that the cutting rate could be improved using this method. However, these approaches have the following disadvantages: (1) additional auxiliary equipment and complex control are needed, (2) the application scope is limited, and (3) these systems involve high energy consumption and high costs.

The third category is to improve the dynamic stiffness of the boring bar. Using high-strength composite materials, many researchers have tried to manufacture boring bars that could improve the dynamic stiffness and achieve large length/diameter (L/D) ratios without chatter. For example, Nagano et al. [21] designed four types of composite boring bars with differently shaped steel cores. Using the finite element method (FEM), they analyzed the performance of these boring bars. The results showed that a boring bar with a cross-shaped steel core exhibited the best cutting capability and stability. In particular, when L/D = 7.0, this kind of boring bar could achieve non-chatter cutting. Lee et al. [22] designed and manufactured a carbon-fiber epoxy composite boring bar. Experimental results showed that the dynamic stiffness of this kind of composite boring bar was about 30% higher than that of a tungsten carbide bar. Furthermore, chatter did not occur when L/D = 10.7. Considering three different supporting conditions, fixed-free, pinned-pinned-free, and spring-spring-free, Ren et al. [23] analyzed the dynamic characteristics and stabilities of composite boring bars. However, they did not consider the shear deformation and rotational inertia in analysis. According to this research, we can conclude that, using a kind of composite material to construct a boring bar for cutting does not require complicated control structures and is more practical than the first two categories. Therefore, the manufacture of composite boring bar is very promising.

All the aforementioned studies neglected the influence of the shear deformation and rotational inertia, which should not be ignored in dynamic characteristic analysis of the boring bar (which is a kind of thick beam model). If the influence of shear deformation and rotational inertia is ignored in the calculation of nature frequency, the result will be much larger than the actual value [24]. That will lead to the inaccurate adjustment of the cutting speed and depth in CNC boring, and even destroy the surface finish of machined workpieces. Moreover, because of the high dynamic stiffness of the composite material, the study about the dynamic characteristics of the composite boring bar with shear deformation and rotational inertia is more conducive to improve the understanding about the inherent dynamic characteristics of the composite materials boring bar and to provide guidance for designing

of boring bars. Therefore, this paper focuses on studying the influence of the shear deformation and rotational inertia on the dynamic characteristic of composite boring bar.

In contrast to existing studies, the novel features of this paper are as follows. First, based on the Adomian Modified Decomposition method (AMDM), this paper presents a composite boring bar model considering shear deformation and rotational inertia. Secondly, the mode shapes of the established a mathematical model is compared and analyzed. Thirdly, the natural frequency and the cutting stability is compared comprehensively. In particular, the convergence about the chatter stability of our model is verified. The analysis results drawn in this paper are helpful to improve the understanding about the inherent dynamic characteristics of the composite boring bar, and to provide guidance for designing of boring bars. Moreover, to further improve the surface finish of machined workpieces, accurate adjustment of the cutting speed and depth in CNC boring can be designed based on the results drawn in this paper.

## 2. Modelling of Boring Bar

In modeling the boring bar, it is generally assumed that the boring bar is a cantilever beam, as shown in Figure 1. The length of the boring bar is L. The positions of the fixed and free ends are at z = 0 and z = L, respectively.

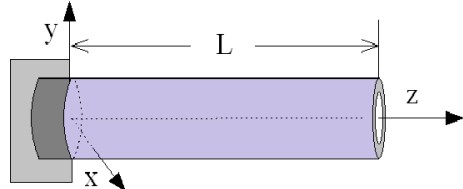

**Figure 1.** Boring bar model.

*2.1. Model of Boring Bar with Shear Deformation and Rotational Inertia*

Based on the AMDM method [25], a model of a boring bar considering shear deformation and rotational inertia can be defined as follows [26–28]:

$$\rho A \frac{\partial^2 y}{\partial t^2} - \kappa A G \left( \frac{\partial^2 y}{\partial z^2} - \frac{\partial \psi}{\partial z} \right) = 0 \tag{1}$$

$$EI \frac{\partial^2 \psi}{\partial z^2} + \kappa A G \left( \frac{\partial y}{\partial z} - \psi \right) - \rho I \frac{\partial^2 \psi}{\partial t^2} = 0 \tag{2}$$

where $\rho$ is the mass density of the boring bar material, $A$ is the cross-sectional area of the boring bar, $\kappa$ the shear correction factor of the boring bar, $G$ represents the shear modulus of the boring bar material, $E$ represents the Young's modulus of the boring bar material, $I$ is the area moment of inertia of the boring bar, and $y = y(z,t)$ and $\psi = \psi(z,t)$ are the transverse deflection of the boring bar and the angle of rotation of the cross-section due to the bending of the boring bar at position $z$ and time $t$, respectively.

According to Figure 1, the boundary conditions at z = 0 (i.e., the fixed end, at which the position, transverse deflection, and angle of rotation are zero) are defined as follows:

$$\kappa A G \left[ \frac{\partial y(z,t)}{\partial z} - \psi(z,t) \right] - k_{TL} y(z,t) = 0 \tag{3}$$

$$EI \frac{\partial \psi(z,t)}{\partial z} - k_{RL} \psi(z,t) = 0 \tag{4}$$

The boundary conditions at z = L (i.e., the free end, at which the position, bending moment, and shear force are zero) are defined as follows:

$$\kappa AG\left[\frac{\partial y(z,t)}{\partial z} - \psi(z,t)\right] = 0 \tag{5}$$

$$EI\frac{\partial \psi(z,t)}{\partial z} = 0 \tag{6}$$

With the following definitions, Equations (1) and (2) can be written in matrix form as Equation (8).

$$X = \frac{z}{l}, \; Y(Z) = \frac{Y(z)}{l}, \; \Psi(Z) = \psi(z), \; \Omega^2 = \frac{\rho A \omega_s^2 l^4}{EI}, \; \eta = \frac{I}{Al^2}, \; \xi = \frac{\kappa GAl^2}{EI}, \tag{7}$$

$$\begin{bmatrix} Y''(Z) \\ \Psi''(Z) \end{bmatrix} + \begin{bmatrix} 0 & -1 \\ \xi & 0 \end{bmatrix}\begin{bmatrix} Y'(Z) \\ \Psi'(Z) \end{bmatrix} + \begin{bmatrix} \frac{\Omega^2}{\xi} & 0 \\ 0 & -\xi + \eta\Omega^2 \end{bmatrix}\begin{bmatrix} Y(Z) \\ \Psi(Z) \end{bmatrix} = \begin{bmatrix} 0 \\ 0 \end{bmatrix} \tag{8}$$

Equation (8) can be simplified as follows:

$$u''(Z) + Bu'(Z) + Du(Z) = 0 \tag{9}$$

where $u(Z) = \begin{bmatrix} Y(Z) \\ \Psi(Z) \end{bmatrix}$, $u'(Z) = \begin{bmatrix} Y'(Z) \\ \Psi'(Z) \end{bmatrix}$, $u''(Z) = \begin{bmatrix} Y''(Z) \\ \Psi''(Z) \end{bmatrix}$, $B = \begin{bmatrix} 0 & -1 \\ \xi & 0 \end{bmatrix}$,

and $D = \begin{bmatrix} \frac{\Omega^2}{\xi} & 0 \\ 0 & \eta\Omega^2 - \xi \end{bmatrix}$.

According to the AMDM [28] solution of Equation (9) can be approximated by the first $k$ terms, as follows:

$$U(Z) = \begin{bmatrix} Y(Z) \\ \Psi(Z) \end{bmatrix} = \begin{bmatrix} \sum_{j=0}^{k-1} C_{1,j}Z^j \\ \sum_{j=0}^{k-1} C_{2,j}Z^j \end{bmatrix} = \sum_{j=0}^{k-1} C_j Z^j = \Phi(Z) + \sum_{j=2}^{k-1} C_j Z^j \tag{10}$$

where:

$$\Phi(Z) = C_0 + C_1 Z = U(0) + U'(0)Z \tag{11}$$

$$C_0 = U(0) = \begin{bmatrix} Y(0) \\ \Psi(0) \end{bmatrix} \tag{12}$$

$$C_1 = U'(0) = \begin{bmatrix} Y'(0) \\ \Psi'(0) \end{bmatrix} \tag{13}$$

$$C_j = \frac{-1}{j(j-1)}\left[(j-1)BC_{j-1} + DC_{j-2}\right], \; j = 2,3,4,.... \tag{14}$$

The boundary conditions of Equations (3) and (4) at z = 0 can be written in matrix form as follows:

$$\begin{bmatrix} \xi & 0 \\ 0 & 1 \end{bmatrix}\begin{bmatrix} Y'(Z) \\ \Psi'(Z) \end{bmatrix} + \begin{bmatrix} -K_{TL} & -\xi \\ 0 & -K_{RL} \end{bmatrix}\begin{bmatrix} Y(Z) \\ \Psi(Z) \end{bmatrix} = \begin{bmatrix} 0 \\ 0 \end{bmatrix} \tag{15}$$

At the fixed end, $Y = 0$ and $\psi = 0$, and the vectors $C_0$ and $C_1$ in Equations (12) and (13) can be rewritten as follows:

$$C_0 = U(0) = \begin{bmatrix} 0 \\ 0 \end{bmatrix} \tag{16}$$

$$C_1 = U(0) = \begin{bmatrix} a_1 \\ a_2 \end{bmatrix} \tag{17}$$

Similarly, the boundary conditions at the free end (z = L) can be written in matrix form as follows:

$$\begin{bmatrix} Y'(Z) \\ \Psi'(Z) \end{bmatrix} + \begin{bmatrix} 0 & -1 \\ 0 & 0 \end{bmatrix}\begin{bmatrix} Y(Z) \\ \Psi(Z) \end{bmatrix} = \begin{bmatrix} 0 \\ 0 \end{bmatrix} \tag{18}$$

where $Y' = \psi$ and $\psi' = 0$.

Taking the derivative of Equation (11) yields the following:

$$U'(Z) = \sum_{j=0}^{\infty} (j+1)C_{j+1}Z^j = C_1 + (j+1)\sum_{j=1}^{\infty} C_{j+1}Z^j \tag{19}$$

Substituting Equations (11) and (19) into Equation (18) yields the following:

$$F^{[k]}(\Omega) = U'(1) + \begin{bmatrix} 0 & -1 \\ 0 & 0 \end{bmatrix} U(1) \tag{20}$$

Extracting the variables $a_1$ and $a_2$ from Equation (20) yields the following:

$$F^{[k]}(\Omega) = f^{[k]}(\Omega) \begin{bmatrix} a_1 \\ a_2 \end{bmatrix} \tag{21}$$

According to Cramer's rule, Equation (21) can be written as

$$\left| f^{[k]}(\Omega) \right| = 0 \tag{22}$$

When Equation (22) is satisfied, the following equation is obtained, and the iterative index $k$ can be calculated:

$$\left| \Omega^{[k]} - \Omega^{[k-1]} \right| \leq \varepsilon \tag{23}$$

Substituting $\Omega$ into the natural frequency function given by Equation (7), we obtain the following:

$$\omega_s = \frac{\Omega}{L^2} \sqrt{\frac{EI}{\rho A}} \tag{24}$$

By substituting $\Omega$ into Equation (10) and making the result dimensionless, the normalized vibration mode function is as follows:

$$\overline{u}(Z) = \begin{bmatrix} Y^{[k]}(Z) / \sqrt{\int_0^l [Y^{[k]}(Z)]^2 dZ} \\ \Psi^{[k]}(Z) / \sqrt{\int_0^l [\Psi^{[k]}(Z)]^2 dZ} \end{bmatrix} = \begin{bmatrix} \overline{Y}^{[k]}(Z) \\ \overline{\Psi}^{[k]}(Z) \end{bmatrix} \tag{25}$$

### 2.2. Boring Bar without Shear Deformation and Rotational Inertia

According to the uniform Euler-Bernoulli beam theory, the differential equation of motion is as follows [29]:

$$EI\frac{\partial^4 y}{\partial z^4} + \rho A \frac{\partial^2 y}{\partial t^2} = 0 \tag{26}$$

The natural frequency of the boring bar is defined as follows [30]:

$$\omega_i = (\lambda_i L)^2 \sqrt{\frac{EI}{\rho A L^4}} \quad (i = 1, 2). \tag{27}$$

The vibration mode function of the boring bar is defined as:

$$Y_{(i)c}(z) = C_1 \left( (\cosh \lambda_i z - \cos \lambda_i z) - \frac{(\sinh \lambda_i L - \sin \lambda_i L)}{(\cosh \lambda_i L + \cos \lambda_i L)} (\sinh \lambda_i z - \sin \lambda_i z) \right) \tag{28}$$

Based on the above equations, we can obtain the first four order values of $\lambda_i L$ in the non-shear boring bar model, which are listed in Table 1, where $\Omega_i$ is the ith estimated dimensionless natural frequency.

**Table 1.** $\lambda_i L$ and $\Omega_i$ for non-shear boring bar and shear boring bar, respectively.

| $i$ | 1 | 2 | 3 | 4 |
|---|---|---|---|---|
| $\lambda_i L$ | 1.8751 | 4.6941 | 7.8548 | 10.9955 |
| $\Omega_i$ | 3.1914 | 14.026 | 30.066 | 46.018 |

### 2.3. Mechanical Calculation of Composite Boring Bars

The composite boring bar is often a stack of multiple layers. The material of each layer can be chosen to have different ply angles based on various requirements. The relationships between the material axis coordinate system 1, 2, 3 and the boring bar coordinate system X, Y, Z are represented in Figure 2 [31].

In Figure 2, $\theta$ represents the angle between the material axis 1 and the Z axis of the boring bar coordinate system, which is named the ply angle. According to the literature [32], the relationship between the stress and deformation of a single-layer composite material in a cylindrical coordinate system and the boring bar composite with many different ply angle layers can be obtained as follows:

$$A = \pi \sum_{k=1}^{N} \left[ R_{o(k)}^2 - R_{i(k)}^2 \right] \tag{29}$$

$$I = \frac{\pi}{4} \sum_{k=1}^{N} \left[ R_{o(k)}^4 - R_{i(k)}^4 \right] \tag{30}$$

$$C_b = EI = \frac{\pi}{4} \sum_{k=1}^{N} \overline{Q}_{11}^{(k)} \left[ R_{o(k)}^4 - R_{i(k)}^4 \right] \tag{31}$$

$$C_s = \kappa GA = \frac{1}{2 - G_{12}v_{12}/E_{11}} G_{12}A \tag{32}$$

where $A$ is the cross-sectional area, $I$ the area moment of inertia, $C_b$ is the bending rigidity, and $C_s$ is the shear rigidity. The above equations are for a composite boring bar. The coefficients $EI$ and $\kappa GA$ of Equations (1) and (2) will be replaced by $C_b$ and $C_s$ respectively. $R_{o(k)}$ and $R_{i(k)}$ are the outer and inner radii of the $k$-th layer of the composite material, respectively, $G_{12}$ is the major shear modulus, $v_{12}$ the major Poisson's ratio, and $\overline{Q}_{11}$ the off-axis elastic stiffness coefficient of the orthotropic materials.

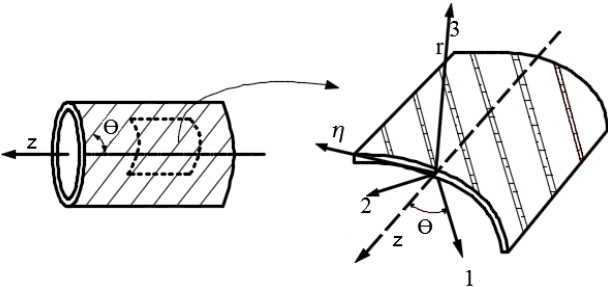

**Figure 2.** Relationship between the material principal direction 1 and the Z axis of boring bar coordinate system.

## 3. Analysis of Vibration Mode Shapes

The vibration mode shapes of the shear boring bar are compared with those of the non-shear boring bar. A carbon composite material was selected as the material to construct the boring bar. The macroscopic mechanical parameters of the carbon composite are shown in Table 2. The boring bar length L was set to 0.6 m, the inside radius $R_i$ was set to 0.043 m, and the outside radius $R_o$ was set to

0.045 m. The ply mode was set to the same angle and contained eight layers (denoted as $[\theta]_8$). The ply angle $\theta$ was set to 45°. The thickness of each layer was set to 0.25 mm.

**Table 2.** Macroscopic mechanical parameters of composite materials.

| Parameters | $E_1$ (Gpa) | $E_2$ (Gpa) | $G_{12}$ (Gpa) | $G_{23}$ (Gpa) | $\nu_{12}$ | $\nu_{23}$ | $\rho$ (kg/m$^3$) |
|---|---|---|---|---|---|---|---|
| Carbon | 181.0 | 10.3 | 7.17 | 3.78 | 0.28 | 0.30 | 1.76 |
| Aramid | 76 | 11.5 | 6.3 | 4.4 | 0.30 | 0.34 | 1.44 |
| Graphite | 25.8 | 8.7 | 3.5 | 3.5 | 0.34 | 0.32 | 1.672 |

According to Equation (23), after 14 iterations, the accuracy was adequate. The result was $\Omega_1 = 3.379317$. The accuracy was calculated as follows:

$$\left|\Omega_1^{14} - \Omega_1^{13}\right| = 0.000008 \le 0.00001 \tag{33}$$

By substituting $\Omega_1$ into Equation (25), one can obtain Equation (34). To meet the accuracy requirement, we selected the first 14 terms to compose the modal function, as follows:

$$
\begin{aligned}
\overline{Y}_1^{14}(Z) =\ & 0.35497Z + 2.93914Z^2 - 1.39423Z^3 - 0.11458Z^4 + 6.27048 \times 10^{-2}Z^5 \\
& + 8.48415 \times 10^{-2}Z^6 - 1.75834 \times 10^{-2}Z^7 - 1.40259 \times 10^{-3} \times Z^8 + 3.25191 \times 10^{-4}Z^9 \\
& + 1.78538 \times 10^{-4}Z^{10} - 2.39683 \times 10^{-5}Z^{11} - 1.83380 \times 10^{-6}Z^{12} + 2.64660 \times 10^{-7}Z^{13} \\
& + 8.03156 \times 10^{-8}Z^{14}
\end{aligned} \tag{34}
$$

Under the same settings of parameters (boring bar length L, radius $R_i$ and $R_o$, ply angle $\theta$, etc.), the vibration mode function of non-shear boring bar is calculated by Equation (28). Therefore, based on the Equations (28) and (34), we plotted the mode shapes of the non-shear boring bar and shear boring bar using the MATLAB software, respectively. The four vibration mode shapes are shown in Figure 3.

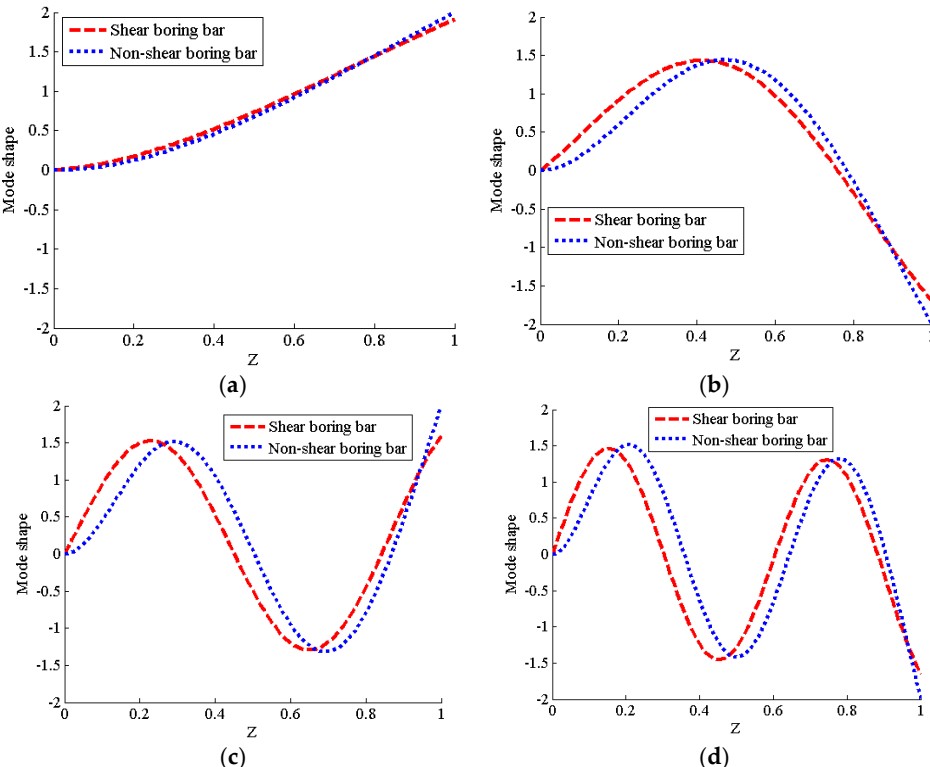

**Figure 3.** Four vibration mode shapes of the shear boring bar and non-shear boring bar. (**a**) First-order mode shape; (**b**) Second-order mode shape; (**c**) Third-order mode shape; (**d**) Fourth-order mode shape.

Figure 3 shows that the first vibration mode shapes of the shear boring bar almost overlapped with the first-order non-shear mode shape of the boring bar. However, for the second-, third-, and fourth-order modes, the spacing between the curves of the two kinds of boring bars increased gradually, and the curves no longer coincided. Thus, it can be concluded that the vibration mode shapes of the boring bar are influenced by the shear deformation and rotational inertia.

## 4. Analysis of the Natural Frequency of Two Boring Bar Models

To complete boring work, the different transverse stiffness of the composite boring bars can be designed according to different boring environments. The following questions remain: How do the shear deformation and rotational inertia affect the boring bar natural frequency? What are the differences between them? Answering these questions will be beneficial for providing theoretical guidance for designing a boring bar.

### 4.1. Comparison of Natural Frequency for Different L/D Ratios

To compare the natural frequencies of the shear boring bar and non-shear boring bar for different L/D ratios, we fixed the inner radius $R_i$ and the outer radius $R_o$ of the boring bar at 0.043 and 0.045 m, respectively. Moreover, based on the same stacking sequences $[\theta]_8$, the ply angle $\theta$ was set with values ranging from $-90°$ to $90°$. The length of the boring bar was set to 0.4, 0.6, and 0.8 m. The value of $\omega_s$ of the shear boring bar was calculated according to the AMDM method. The value of $\omega$ of the non-shear boring bar was calculated using Equation (27). Based on these two calculation methods, we use the MATLAB software to calculate the variation of the natural frequencies of the shear and the non-shear boring bars. The resulting values of $\omega$ are shown in Figure 4, in which "L/D" with subscript "S" denotes the shear boring bar values and "L/D" without the subscript "S" denotes the non-shear boring bar values.

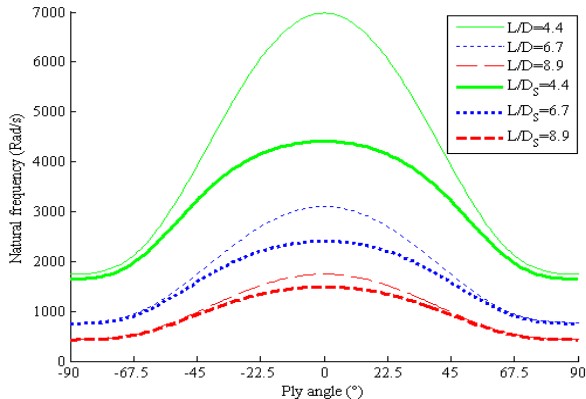

**Figure 4.** Comparison of natural frequencies for different L/D ratios.

Figure 4 shows that the natural frequency curves of the two kinds of boring bar models are symmetrically distributed on both sides of the ply angle of $0°$. The natural frequency curves are different for different L/D ratios. For the same bar, a smaller L/D ratio (L/D = 4.4) yields a larger natural frequency. The larger the ratio is, the lower the natural frequency becomes. Moreover, comparing the two composite boring bar models, the natural frequency curves of the shear boring bar model are below the corresponding non-shear boring bar curves for the same L/D ratio. The closer the ply angle to $0°$, the greater the natural frequency difference between them is. However, if the L/D ratio is large and the ply angle is near $\pm90°$, the frequencies of the two models are almost equal to each other. Therefore, we can conclude that, when the influence of the shear deformation and rotational inertia are considered, the natural frequency of the shear boring bar model is reduced. The smaller the L/D ratio of the composite boring bar, the smaller the ply angle is, and the greater the influence of the shear deformation on the natural frequency is.

### 4.2. Comparison of Natural Frequencies at Different T/D Ratios

To compare the natural frequencies of the two models for different T/D ratios, the length of the boring bar was set to 0.6 m, and the number of layers was 4, 8, and 12. Moreover, the T/D ratio was set to values of 0.0111, 0.0222, and 0.0333, respectively. The other parameters were set with same values as those in Section 4.1. The first-order natural frequency curves of these two kinds of boring bar models under different ply angles and different T/D ratios are shown in Figure 5.

Figure 5 shows that the natural frequency curves of these two boring bar models are symmetrically distributed about 0°. The natural frequency curves for different T/D ratios are different. The natural frequency curve of the boring bar with large T/D ratio (e.g., 0.0333) is at the top, whereas the natural frequency curve of the boring bar with a small T/D ratio is at the bottom. Comparing the natural frequencies of the two kinds of boring bar models at the same T/D ratio, the frequency curves of the model without shear deformation and rotational inertia included are at the top, whereas the shear boring bar model curves are at the bottom. In particular, the two frequencies are quite different when the ply angle ranges from −45° to 45°. In the ranges of [−90°, −45°] and [45°, 90°], the frequency curves of these two models almost overlap. Thus, we can conclude that the larger the T/D ratio of the composite boring bar, the smaller the laying angle is, and the greater the influence of the shear deformation and rotational inertia on the natural frequency is.

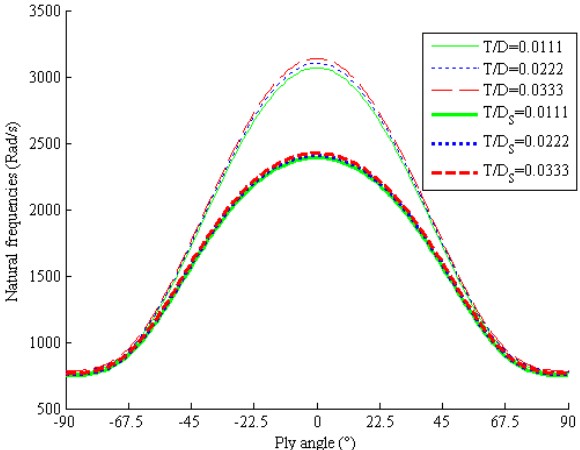

**Figure 5.** Natural frequencies of the two models for different T/D ratios.

### 4.3. Comparison of Natural Frequencies for Different Composite Materials

To compare the natural frequencies of these two models with different composite materials, three kinds of composite materials were selected: carbon, aramid, and graphite composites.

These three kinds of materials are selected because they are popular [21,22,33] in making boring bars. The macroscopic mechanical parameters of the composite materials are shown in Table 2. The length of the composite boring bar was 0.6 m, the thickness of the boring bar was 2 mm, and the ply angle was 45°. The natural frequencies of these two models for different ply angles and different composite materials are shown in Figure 6.

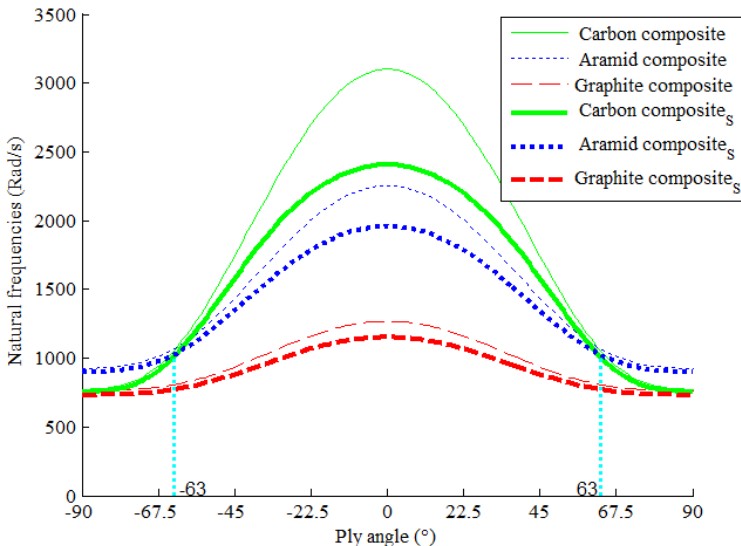

**Figure 6.** Natural frequencies of these two models for different composite materials.

Figure 6 shows that regardless of which composite materials were used, the natural frequency curves of these two models are symmetrically distributed at a ply angle of 0°. The natural frequency curves are different for the different composites. Moreover, the natural frequency curve of carbon composite boring bar is at the top, followed by the aramid composite and graphite composite curves. The two curves of the aramid and carbon composites intersect at the symmetric positions of 63° and −63°. The natural frequencies of the carbon composite are higher than that of the aramid composite when the absolute value of the ply angle is less than 63°. Comparing the natural frequencies of the two kinds of boring bar models composed of the same composite, the frequency curves of the model without shear deformation and rotational inertia are at the top, whereas the shear boring bar model curves are at the bottom. In addition, the two models almost overlap in the ply angle ranges of [−90°, −63°] and [63°, 90°].

## 5. Analysis of Chatter Stability of Two Boring Bar Models

To analyze how the shear deformation and rotational inertia affect the chatter stability of the boring bar models, we designed a one-dimensional model to study the critical cutting depth and the corresponding spindile speed, which are calculated according to Equations (31) and (32) [34]:

$$b_{\text{lim}} = \frac{2\zeta\omega_1\omega_c M}{K_c Y_1^2(L)\sin\left(2\arctan\left(\frac{2\zeta\omega_1\omega_c}{\omega_c^2-\omega_1^2}\right)\right)} \tag{35}$$

$$n = \frac{60\omega_c}{(2k+1)\pi + 2\arctan\left(\frac{2\zeta\omega_1\omega_c}{\omega_c^2-\omega_1^2}\right)}, k = 0, 1, 2 \tag{36}$$

where $b_{\text{lim}}$ is the cutting depth, $n$ is the spindle speed, $\omega_c$ is the chatter frequency of the boring bar, $\omega_1$ is the first natural frequency of the boring bar, $\zeta$ is the damping coefficient of the boring bar, $M = \int_0^L \rho A Y_i^2(z)dz$ is the mass of the boring bar, and $K_c$ is the coefficient of the cutting force (N/mm$^2$). In order to compare the difference of cutting stability of these two boring bars, the damping coefficient $\zeta$ is set to 0.1358 and the cutting force coefficient $K_c$ is set to $2.3391 \times 10^6$, and other parameters are set according to the comparison requirements (which is illustrated in the following subsections). The natural frequency $\omega_1$ of shear boring bar is calculated by AMDM method. Differently, the natural frequency of non-shear boring bar is calculated by Equation (27). The MATLAB software is used to plot Lobe diagram.

*5.1. Convergence of the Chatter Stability*

As discussed in Section 4, the natural frequency of the shear boring bar was solved using the AMDM method. The result was an approximate value that was calculated by iteration of the first $k$ terms. In Section 3, the accuracy of the $\Omega_1$ value was one-ten thousandth. Whether the value of $\Omega_1$ affects the chatter stability of the boring bar and the reliability of the $\Omega_1$ result must be tested by experiments.

We first selected an iteration number $k = 16$ to verify whether the resulting $\Omega_1$ value was within the desired tolerance of $10^{-6}$. The test result showed that when $k = 16$, the obtained $\Omega_1$ value no longer changed. Therefore, $\Omega_1$ achieved a stable value, which was chosen as the reference standard. To further verify whether the $\Omega_1$ value was stable when $k$ was assigned values smaller than 16, we calculated the results for four other iteration numbers ($k = 7, 9, 11$, and 13). Table 3 lists the iteration number, the related $\Omega_1$ values, and the ultimate cutting depth.

The chatter stability lobes are shown in Figure 7. The five chatter stability lobes converged to two curves. The chatter stability lobe when $k = 7$ was different than the others, as it did not converge to the standard curve (i.e., the curve when $k = 16$). When $k = 9$, the chatter stability lobe basically overlapped the $k = 16$ curve. When $k = 11$ and 13, the chatter stability lobes overlapped and could not be distinguished from the reference curve.

**Table 3.** Five values of different iteration number.

| Iterative Number | $k = 7$ | $k = 9$ | $k = 11$ | $k = 13$ | $k = 16$ |
|---|---|---|---|---|---|
| $\Omega_1$ Value | 3.2558831 | 3.1803316 | 3.1915264 | 3.1913896 | 3.1913967 |
| Ultimate cutting depth ($V_k$) | 0.20093313 | 0.19199067 | 0.19276827 | 0.19315708 | 0.19315708 |
| $\lvert V_k - V_{16} \rvert$ | 0.0078 | 0.0012 | $3.8881 \times 10^{-4}$ | 0 | 0 |

To distinguish these overlapping curves, we calculated the ultimate cutting depths for each iteration number, as shown in Table 3. The ultimate cutting depth difference was 0.0078 when $k = 7$, 0.0012 when $k = 9$, and $3.8881 \times 10^{-4}$ when $k = 11$. In particular, when $k = 11$, the cutting depth difference met the accuracy requirement of $10^{-4}$. When $k = 13$, the cutting depth difference was 0, which completely satisfied the accuracy requirement. Therefore, the results listed in Table 3 and Figure 7 is consistent with the analysis results obtained in Section 3.

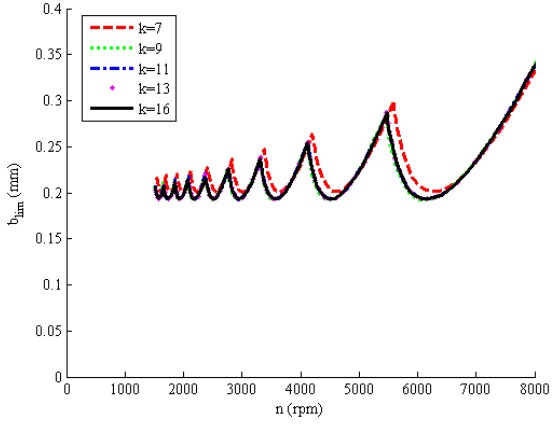

**Figure 7.** Chatter stability for different iteration numbers.

*5.2. Comparison of Chatter Stability with Different Ply Angles*

To compare the chatter stabilities of these two kinds of composite boring bar models under different ply angles, the carbon composite was selected as the composite material. According to the experimental results shown in Section 4.3, the two boring bar models for the carbon composite yielded

the highest natural frequency. Similar to the above experiments, the lengths of the boring bars were 0.6 m, the diameters were 0.09 m, eight layers were included, and each layer thickness was 0.25 mm. The ply angles were set to 0°, 45°, and 90°. The chatter stability lobes of the two models are shown in Figure 7, where those of the shear boring bar model are labeled with the subscript S.

Figure 8 shows that the chatter stability lobes of the two boring bar models differed at different ply angles. The chatter stability lobe is at the top when the ply angle is 0°, and at the bottom when the ply angle is 90°. When the ply angle is 45°, the curve falls in between the other curves. Thus, the smaller the ply angle is, the deeper the cut is. Moreover, with the same ply angle, the chatter stability lobes of the non-shear boring bar model are on top, whereas those of the shear boring bar model are below. The cutting depth is reduced when shear deformation and rotational inertia are considered. In particular, when the ply angle is 0°, the difference between these two models is the greatest. As the ply angle increases, the difference becomes smaller and the chatter stability lobes almost overlap.

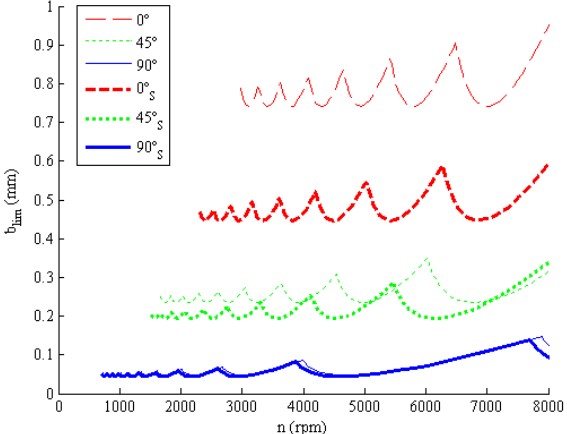

**Figure 8.** Chatter stability lobes of the two models with different ply angles.

### 5.3. Comparison of Chatter Stability with Different Stacking Sequences

To analyze the influence of different stacking sequences on the chatter stability of the two boring bar models, we designed six stacking sequences (see Table 4). Each sequence contained eight layers. In the first two sequences, there were two 0° layers, two 45° layers, and four 90° layers. The latter four sequences included four 0° layers, two 45° layers, and two 90° layers. The chatter stabilities of the two boring bar models under these six stacking sequences were calculated. The resulting lobe curves are drawn and shown in Figure 9.

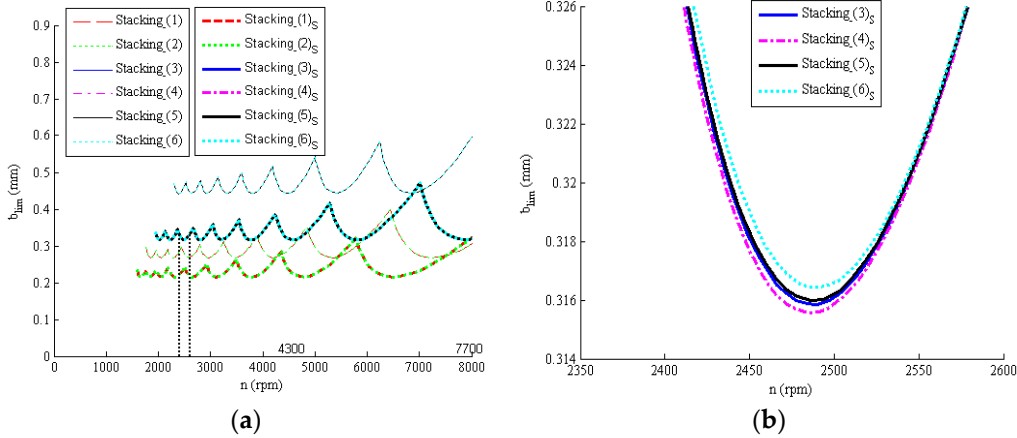

**Figure 9.** Chatter stability lobes with different stacking sequences. (**a**) Chatter stability lobes of six typical sequences; (**b**) Chatter stability lobes of the latter four sequences.

**Table 4.** Comparison of the ultimate cutting depth of composite boring bars.

| No. | Stacking Sequence | Ultimate Cutting Depth | |
|---|---|---|---|
| | | Non-Shear Boring Bar (mm) | Shear Boring Bar (mm) |
| 1 | $[90_2, 45, 0]_s$ | 0.2667 | 0.2148 |
| 2 | $[90, 0, 90, 45, 90, 45, 0, 90]$ | 0.2673 | 0.2151 |
| 3 | $[90, 45, 0_2]_s$ | 0.4402 | 0.3158 |
| 4 | $[0_2, 45_2, 90_2, 0_2]$ | 0.4399 | 0.3156 |
| 5 | $[0_2, 90, 45]_s$ | 0.4406 | 0.3160 |
| 6 | $[45, 0, 45, 0, 90, 0, 90, 0]$ | 0.4413 | 0.3164 |

\* "s" means symmetrical stacking. "2" refers to stacking two layers at the same ply angle.

Figure 9a shows four curves. Under the same boring bar model, the differences between the chatter stability lobes of sequences 1 and 2 were very small. Moreover, the differences between the chatter stability lobes of sequences 3–6 were also relatively small. Figure 9b shows an enlarged section (section highlighted by dashed rectangle in Figure 9a of the 3rd–6th sequences of the lobe curves for the shear boring bar in Figure 9a.

In Figure 9a, the chatter stability lobes of the two boring bar models are different for the different stacking sequences. The chatter stability curves of the 1st–2nd sequences (including four 90° layers) are below, and the chatter stability curves of 3rd–6th sequences (including four 0° layers) are above. The greater the number of 0° layers was, the greater the cutting depth was. Furthermore, the chatter stability lobes of the non-shear boring bar model are at the top, whereas the chatter stability lobes of the shear boring bar model are at the bottom. These results illustrated that the cutting depth decreased when the shear deformation and rotational inertia were considered. As shown in Figure 9b, though all the 3rd, 4th, 5th, and 6th stacking sequences included the same number of layers (i.e., eight layers), the cutting depths of the composite boring bars are different. The best chatter stability lobe was obtained by the 6th sequence, followed by the 5th sequence, the 3rd sequence, and the 4th sequence.

The ultimate cutting depth in Figure 9a was obtained and is listed in Table 4. As shown in Table 4, the ultimate cutting depths of the shear boring bar model were significantly lower than those of the non-shear boring bar model. Comparing the ultimate cutting depths of the 1st and 2nd sequences, the value of the 2nd sequence was larger (i.e., better). Similarly, for stacking sequences 3–6, the best ultimate cutting depth was for the 6th sequence, which is consistent with the result shown in Figure 9b.

### 5.4. Comparison of Chatter Stability for Different L/D Ratios

To compare the chatter stabilities of two boring bar models for different L/D ratios, we set the lengths of carbon composite boring bar (i.e., L) to 0.4, 0.6, and 0.8 m. The L/D ratios of the boring bar were 4.4, 6.7, and 8.9, respectively. The ply angle was set to 45°, and other parameters, such as the inside and outside radii, were assigned with similar values to those in Section 5.2. The chatter stability lobes of these two kinds of boring bar models are shown in Figure 10.

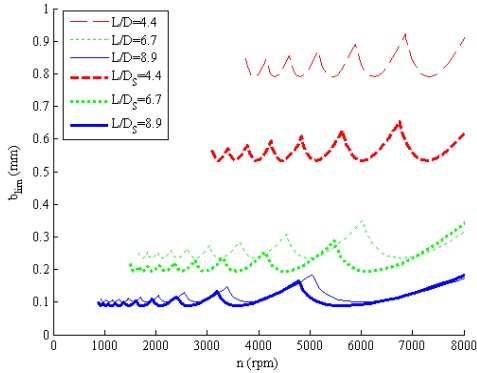

**Figure 10.** Chatter stability lobes for different L/D ratios.

Figure 10 shows that the chatter stability lobes of the two composite boring bar models were different for different L/D ratios. The lobe curve for the L/D ratio of 4.4 is at the top, the lobe curve for the L/D ratio of 8.9 is at the bottom, and the lobe curve for L/D = 6.7 is in the middle. Moreover, the chatter stability curve of the shear boring bar model (which is labeled with the subscript S) is lower than that of the non-shear boring bar model when the shear deformation and rotational inertia are taken into account. In particular, the larger the L/D ratio is, the smaller the difference between the chatter stability lobes of the shear model and the non-shear model become. When the L/D ratio is 8.9, the chatter stability lobes of the shear model and the non-shear model almost overlap. Thus, it can be concluded that the smaller the L/D ratio of boring bar, the greater the influence of shear deformation and rotational inertia on the chatter stability of boring bar is.

### 5.5. Comparison of Chatter Stability for Different T/D Ratios

To compare the chatter stabilities of these two models for different T/D ratios, the number of layers was set to 4, 8, and 12. The corresponding T/D ratios were 0.0111, 0.0222, and 0.0333, respectively. Other parameters, such as the boring bar length, inside radius, and composite type (carbon), and ply angles were set with similar values to those in Section 5.2. The chatter stability lobes of these two composite boring bar models are shown in Figure 11.

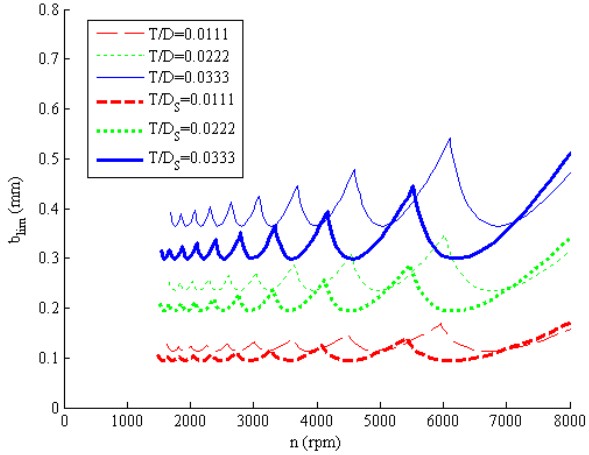

**Figure 11.** Chatter stability lobes for different T/D ratios.

As shown in Figure 11, the chatter stability lobes of the composite boring bars for different T/D ratios were different. The chatter stability lobe lies at the top when T/D = 0.0333, and the chatter stability lobe lies at the bottom when T/D = 0.0111. When T/D = 0.0222, the curve lies in the middle position. The bigger the T/D ratio is, the deeper the cut is. Moreover, with the same L/D ratio, the curves of the shear model (which are labeled with subscript S) always lie below the non-shear boring bar curves. Thus, the cutting depth of the boring bar decreases when considering the shear deformation and rotational inertia. In particular, when T/D = 0.0333, the difference between these two models is the greatest. When T/D = 0.0111, the difference between these two models is a minimum. Therefore, it can be concluded that the greater the T/D ratio, the greater the influence of shear deformation and rotational inertia on cutting stability is.

### 5.6. Comparison of Chatter Stability with Different Composites

To compare the chatter stabilities of the two boring bar models, three different composites were selected: carbon, aramid, and graphite composites. The length of the boring bar was set to 0.6 m, eight layers were laid, and the ply angle was set to 45°. Other parameters, such as the inside and outside radii and thickness of boring bar, were similar to those listed in Section 5.2. The resulting chatter stability lobes of these two boring bar models are shown in Figure 12.

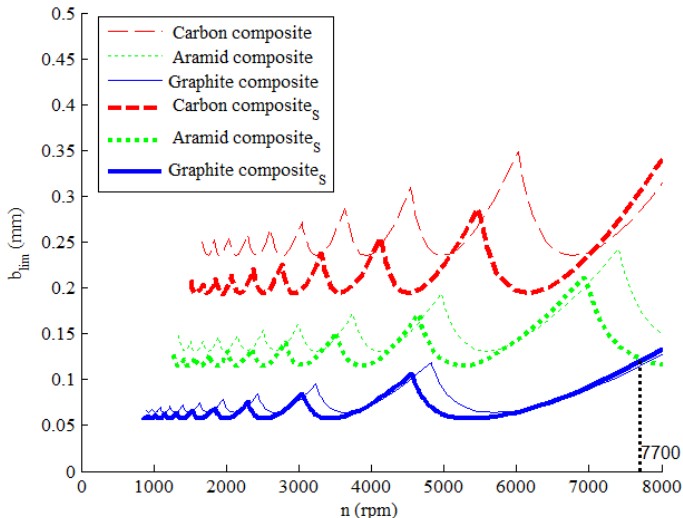

**Figure 12.** Chatter stability lobes of different composite boring bars.

As shown in Figure 12, the chatter stability lobes of the carbon composite are on the top, those of the graphite composite are on the bottom, and those of the aramid are in the middle. Thus, the cutting depth of the carbon composite is better than those of the other composites. Moreover, the chatter stability lobe of the shear boring bar model (which is labeled with the subscript S) is lower than that of non-shear boring bar model when the shear deformation and rotational inertia are taken into account. Thus, the cutting depth of the shear model decreases with the consideration of the shear deformation and rotational inertia. In particular, the chatter stability lobe of the graphite composite intersects with that of the aramid composite at rotation speeds 7700 rpm. Therefore, regardless of composite materials used, the boring depth may be the same.

## 6. Conclusions

In this paper, a shear composite boring bar was modeled and comprehensively compared with a non-shear composite boring bar. The results were as follows.

(1) The vibration modal shapes of the shear and the non-shear composite boring bars are not overlapped. Moreover, the difference between the modal shape curves of the high modal shapes (second-, third-, and fourth-order shape) increased gradually with the increase in the order.
(2) The natural frequencies of these two boring bar models are affected by the ply angles, L/D ratios, the T/D ratios, and type of composite materials. The larger the L/D ratios, the smaller the T/D ratios, or the larger the ply angles, the lower the natural frequency became. The natural frequencies of the shear boring bar are correspondingly lower than those of the non-shear boring bar.
(3) The lobe curves converged when a suitable number of iteration terms are chosen, which prove that the nature frequency that is calculated using the AMDM method is accurate.
(4) The chatter stabilities of these two models were related to the ply angle, stacking sequence, L/D ratio, T/D ratio, and different composite materials. The smaller the ply angle, the smaller the L/D ratio, or the larger T/D ratio, the deeper the cutting depth became. The chatter stability of the shear boring bar model is correspondingly worse than that of the non-shear boring bar model in most cases.
(5) The comparison about the chatter stability of three popular kinds of composite materials show that the carbon composite boring bar is the best one.

This work is a part of the project that focuses on the study of chatter stability in cutting process with composite tooling system. The experimental verification of the presented model using real boring bars will be the subject of a study in the future.

**Author Contributions:** Y.R. proposed the main idea, and the validation method; C.Z. derived the equations in the model, coded the computer program, and obtained the calculations; S.J. wrote the manuscript and drew the figures; J.Z. helped perform the analysis with constructive discussions. All authors have read and agreed to the published version of the manuscript.

**Funding:** This research was funded by the National Science Foundation of China grant number 11672166, and China Postdoctoral Science Foundation Funded Project (2016M5922627165).

**Acknowledgments:** This work was supported by the National Science Foundation of China under contract number 11672166.

**Conflicts of Interest:** The authors declare that there is no conflict of interest regarding the publication of this paper.

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
