# Peer review of "Analysis of the Composite Boring Bar Dynamic Characteristics Considering Shear Deformation and Rotational Inertia"

_applsci, doi:10.3390/app10041533_

Round 1

Reviewer 1 Report

Dear authors, I would like to congratulate you on a very nice paper and interesting research. In my opinion, the paper has only small flaws. Please, consider these changes.

1) The name of the article is not very clear I would recommend to change it a little bit, e.g. to Analysis of the composite boring bar dynamic characteristics considering shear deformation and rotational inertia.

2) Moreover, there is not a chapter Analysis in your paper. Since the word analysis is mentioned already in the name of the paper I would recommend renaming the chapters. Furthermore, please, try to use the IMRAD (IMRaD) structure. I think that it will make the paper more clear and interesting, e.g. lines 108 - 119 could be the beginning of the chapter Methods.

Reviewer 2 Report

The paper is interesting because it concerns the examination of the composite boring bar. A detailed review of literature in section “Introduction” is remarkable.

However, the paper needs small improvements and comments.

In the paper, there are lacks detailed information about the software used for simulation studies. In addition, a more detailed description of the simulation procedures will definitely increase the value of the paper.

Section 4.: What do you mean by term “vibrational mode shape”.

Page 10, line 258: Why carbon, aramid and glass composites were selected for analysis? Are glass composites materials used in the construction of boring bars in practice?

Page 11, lines 305, 306: Is 10-4, it should be 10-4.

Page 5, line 122: How many points were measured?

The conclusions are too general and do not clearly indicate the research goal achieved.

Page 14, line 269: Why in the description of the measuring instrument does not specify how the clearance can be tested?

Moreover, there is no description of the direction of further research. Are the authors planning an experimental verification of their results using real boring bars?
